# Empirical Investigation of the Verbal Cues Involved in Delivering Experiential Metaphors

**DOI:** 10.3390/ijerph182010630

**Published:** 2021-10-11

**Authors:** Eduar S. Ramírez, Francisco J. Ruiz, Andrés Peña-Vargas, Paola A. Bernal

**Affiliations:** Department of Psychology, Fundación Universitaria Konrad Lorenz, Bogotá 110231, Colombia; eduars.ramirezr@konradlorenz.edu.co (E.S.R.); jaimea.penav@konradlorenz.edu.co (A.P.-V.); paolaa.bernalg@konradlorenz.edu.co (P.A.B.)

**Keywords:** metaphor, acceptance and commitment therapy, relational frame theory, deictic relations, relational elaboration

## Abstract

Delivering metaphors experientially has been emphasized in several psychotherapies, such as acceptance and commitment therapy. However, few research has analyzed the variables involved in the efficacy of metaphors. This experimental analog study aims to advance in this topic by analyzing the effect of two components involved in the experiential delivery of metaphors in psychotherapy. The first component is presenting the metaphor by asking the individual to imagine herself as the protagonist of the story versus presenting the metaphor in the third person (Self vs. Other). The second component is the inclusion of verbal cues prompting the relational elaboration of the rules derived from the metaphor content versus not including these prompts (Elaboration vs. No Elaboration). The effect of these components was tested in a double-blind, randomized, 2 × 2 factorial experiment that used the cold pressor task (CPT). Eighty-four participants were exposed to the CPT at the pretest. Afterward, participants were randomly assigned to four experimental protocols. The protocols were audiotaped and consisted of the same metaphor presented in four slightly different ways. Specifically, the protocol of Condition A involved a metaphor with Self and Elaboration, Condition B involved Self and No Elaboration, Condition C involved Other and Elaboration, and Condition D involved Other and No Elaboration. Then, participants were re-exposed to the CPT in the posttest. We hypothesized that Condition A (Self and Elaboration) would show a higher mean increase in pain tolerance than the remaining conditions, which would show similar results. The results were consistent with this hypothesis because Condition A showed a higher percentual increase in pain tolerance (Condition A: *M* = 268.21, *SD* = 167.47; Condition B: *M* = 180.86, *SD* = 73.01; Condition C: *M* = 204.81, *SD* = 100.19; Condition D: *M* = 175.41, *SD* = 76.00). A Bayesian informative hypothesis evaluation showed that this hypothesis obtained the highest posterior model probability. Thus, the results indicate that introducing metaphors by asking the individual to imagine herself as the protagonist of the story and providing prompts for relational elaboration might increase the therapeutic effect of the metaphor.

## 1. Introduction

Psychotherapy usually addresses complex and abstract issues such as intrapersonal conflicts and ineffective strategies to deal with thoughts and emotions. Directly talking about these issues can be difficult for both the therapist and the client. It is not surprising then that multiple forms of psychological interventions use more indirect ways of talking, such as metaphors (e.g., [1,2,3]). In this sense, metaphors allow the therapists to create verbal contexts in which the clients can understand their problems by referring to another domain that is clearer and more representative. Metaphors can also provide the client a vehicle to explore different alternative behaviors and symbolically experiencing their nonarbitrary associated contingencies. This way, metaphors might lead to greater behavioral flexibility than direct instruction by the therapist, which might highlight arbitrary, therapist-mediated consequences [4].

Several psychological intervention models, such as acceptance and commitment therapy (ACT; [5]), have emphasized the relevance of delivering metaphors experientially [6]. Experiential metaphors create a symbolic context in which the client vividly contacts the consequences of his or her behavior. Delivering experiential metaphors can be contrasted with the mere exposition of the metaphor, which might have only a reduced effect because it does not facilitate contact with the emotional functions associated with the consequences of the client’s behavior. Accordingly, the therapist’s skill in delivering experiential metaphors is a critical therapeutic competence in ACT. However, despite the relevance given to experiential metaphors in ACT, scarce research has been conducted regarding the verbal cues that can intensify the client’s experience when delivering metaphors.

The contextual behavioral approach of human language and cognition represented by relational frame theory (RFT; [7]) can provide insights into this issue. Briefly, RFT suggests that relational framing underlies human linguistic and cognitive abilities. Relational framing is an operant behavior consisting of relating stimuli based on arbitrary relational cues such as coordination (“is,” “same as,” etc.), opposition (“opposite to”), comparison (“more than,” “less than”), hierarchy (“includes,” “part of”), causal (“if... then”), deictic (I-You, Here-There, Now-Then), etc. Relational framing allows deriving myriads of multiple untrained relations among stimuli, which will conform relational networks. Importantly, stimulus functions might be transformed according to their relations with the other stimuli of the network. For instance, the functions of an unknown person might change when knowing that he is the best friend of the individual’s worst enemy (i.e., a relation of coordination is established between the unknown person and the individual’s enemy). This way, a neutral stimulus can acquire aversive functions due to the relation established with an aversive stimulus. 

In RFT terms, rules are relational networks in which stimuli are usually framed through different relational cues that might lead to the transformation of the stimulus functions. For instance, the rule “The only way to go to my daughter’s wedding is to cross the ocean” might weaken the avoidant function of the individual’s fear of flying by establishing a conditional relation between flying and attending her daughter’s wedding. Three types of rule-governed behavior have been identified in RFT [7,8,9]. Firstly, pliance is a functional class of rule-following in which the individuals behave according to the rule content because they expect to be reinforced by arbitrary, socially mediated consequences. A specific instance of pliance is called a ply. For instance, an incompetent therapist might instruct a depressed client to engage in behavioral activation to obtain his or her approval as a valid person. This clinical interaction might hinder contact with the natural consequences of engaging in valued actions and would be especially counterproductive with clients displaying generalized pliance, a pattern of rule-governed behavior in which social approval becomes the individual’s primary source of reinforcement. Secondly, tracking is a functional class of rule-following in which the individuals behave according to the rule content because they expect to be reinforced by the natural consequences of the action. A specific instance of tracking is called a track. For example, a therapist might conduct a Socratic dialogue in order for the client to derive and follow a rule specifying the natural consequences of avoiding fear: the fear persists, and the person is not able to engage in valued actions in its presence. Lastly, augmenting is a functional class of rule-following in which the individuals behave according to the rule content because relational networks altered the reinforcing functions of some stimuli contained in the rule. A specific instance of augmenting is called an augmental. Augmenting is rarely seen in its pure form but instead interacting with pliance or tracking [9,10]. For instance, the previous examples of pliance and tracking involved relational networks that altered some stimulus functions; therefore, these rules are also augmentals. 

From the RFT standpoint, analogies and metaphors involve relating two relational networks. This relational activity usually leads to deriving new rules that will transform some stimulus functions [11,12,13]. For example, take into account the swamp metaphor, a typical ACT metaphor in which the individual’s problem is related to a hypothetical scenario in which the only way to advance towards a value is to be willing to experience fear and disgust [5,14]. The metaphor usually takes this form: “It seems that your situation with attending to your daughter’s wedding is similar to the following scenario. Imagine you are at the edge of a big swamp. On the other side of the swamp, there is the most important thing for you. The water of the swamp is filthy and has bugs that scare you a lot. However, the only way to advance towards that thing you care about is to cross it. Would you jump into the water and swim despite the discomfort of the filthy water?” With this metaphor, the ACT therapist aims for the client to derive and follow the track that the only way to attend her daughter’s wedding is to be willing to experience fear during the flight, weakening the fear’s avoidant functions.

The research on the components that increase metaphor efficacy should consider the aim of the therapy [4,15]. According to ACT, the therapy aims to foster psychological flexibility, which can be defined in midlevel terms as the skill of nonjudgmentally contact ongoing private experiences while directing behavior towards valued ends. An RFT conceptualization of psychological flexibility might shed some light regarding the components that enhance the metaphor effect. In RFT terms, psychological flexibility has been conceptualized as the generalized repertoire of framing ongoing behavior in hierarchy with the deictic I, which typically reduces the discriminative functions of ongoing behavior and allows the derivation of rules that specify appetitive augmental functions and behavior in accordance with them [16]. 

Recent research on the components that increase the metaphor effect has used RFT conceptualizations of metaphor and psychological flexibility. For instance, analogies among relational networks containing common physical properties are judged as more apt and are derived more easily than analogies among networks without this characteristic [17]. Sierra, Ruiz, Flórez, Riaño-Hernández, and Luciano [18] used the cold pressor task (CPT) and slightly modified the swamp metaphor to demonstrate that metaphors containing common physical properties with the participants’ pain (i.e., metaphors involving being in contact with cold water) are more effective than the same metaphors without the common physical properties (metaphors involving being in contact with filthy water). Additionally, the study demonstrated that metaphors specifying appetitive augmentals (i.e., values) for being in contact with pain were more efficacious than those that did not specify them. Finally, the study by Criollo, Díaz-Muelle, Ruiz, and García-Martín [19] replicated the finding that common physical properties increase the metaphor effect even in the context of multiple examples of functionally equivalent metaphors. 

This study aims to advance the experimental analysis of the components involved in the experiential delivery of metaphors. Specifically, two components are analyzed in this study. The first component will be called Self vs. Other. “Self” indicates that the metaphor is introduced by asking participants to consider themselves as the protagonist of the metaphor, whereas “Other” indicates that the metaphor is introduced with a fictitious character as the protagonist. Following the RFT approach of psychological flexibility, presenting the metaphor with the client as the protagonist (i.e., Self) might lead to deriving a rule with a more intense transformation of emotional functions than when the metaphor is introduced in the third person (i.e., Other). However, in the latter case, a more intense and personally relevant transformation of functions will depend on the participant’s perspective-taking skills (i.e., fluency in deictic framing).

The second component will be called Elaboration vs. No Elaboration. “Elaboration” involves the inclusion of relational cues to prompt the relational elaboration of the rules derived from the metaphor. This can be contrasted with introducing the metaphor in an expositive way without prompting the elaboration of the rules with questions and/or pauses for the client to derive the consequences (No Elaboration). According to the RFT approach of psychological flexibility, prompting relational elaboration promotes the specification of appetitive augmental functions. In contrast, if relational elaboration is not prompted, the derivation of appetitive augmental functions will depend on the participant’s relational framing skills (i.e., participants with a fluent repertoire in deriving rules might contact the augmental functions, but clients without sufficient fluency would not have the opportunity to contact them).

This study hypothesized that both components (i.e., Self and Elaboration) would show a positive interaction. On the one hand, the differential effect between presenting the metaphor with the participant as the protagonist or by referring to a fictitious character might not be relevant when introducing the metaphor in an expositive way (i.e., No Elaboration). This lack of differential effect would be due to the absence of time to elaborate the rules derived from the metaphor. Conversely, the differential effect might be maximized when prompting relational elaboration of the rules involved in the metaphor. On the other hand, the difference between prompting relational elaboration or not prompting it might be noted mostly when the metaphor is introduced with the client as the protagonist (i.e., the effect of this prompting would not depend on the client’s perspective-taking skills). 

## 2. Materials and Methods

### 2.1. Participants

Eighty-four undergraduates (42 women and 42 men; age range = 18 to 40; M = 19.64, SD = 2.75) attending different courses in a university in Bogotá (Colombia) participated in the experiment. The inclusion criteria were (a) being an undergraduate in the university in which the study was conducted (the Institutional Bioethical Committee asked this to ensure that participants were protected by the institutional insurance in case of accident or an unlikely adverse reaction to the experimental procedures), and (b) being equal or older than 18 years. The exclusion criteria were (a) having previous experience with the procedures (i.e., the CPT) or the theory involved in this study (i.e., RFT and ACT), and (b) suffering from cardiac and circulatory affections, hypertension, diabetes, epilepsy, chronic pain conditions, or recent wounds. The latter exclusion criteria were established because the experimental task might adversely affect participants suffering from these conditions. Participants were remunerated COP 10,000 (approximately USD 3) for completing the study. 

### 2.2. Desing and Variables

This study follows a double-blind, randomized, 2 × 2 factorial design that analyzes the effect of two independent variables: (a) presenting the metaphor by asking the participants to imagine themselves as the protagonists of the story or by imagining that the protagonist is a fictitious person (Self vs. Other), and (b) including relational cues prompting for the relational elaboration of rules that might be derived from the metaphor (Elaboration vs. No Elaboration). The combination of these independent variables led to four experimental conditions. The protocol of Condition A involved a metaphor presented in the first person that included cues prompting relational elaboration (i.e., Self and Elaboration). The metaphor of Condition B was also presented in the first person but did not include cues prompting relational elaboration (Self and No Elaboration). The metaphor of Condition C was presented in the third person and included cues prompting relational elaboration (Other and Elaboration). Lastly, the metaphor in Condition D was presented in the third person and did not include cues prompting relational elaboration (i.e., Other and No Elaboration).

The primary dependent variable was pain tolerance as measured by the percentage of time tolerating the pain at posttest in relation to pretest, where spending the same amount of time at posttest as at pretest would correspond to 100%, spending twice the time would be 200%, and so on. These scores were computed because performance in the CPT is known for substantial variability across participants; thus, scoring performance this way helps to control tolerance time at pretest. Differential scores can be tricky in this case because a difference of 10 s is not the same for a participant who tolerated the task for 100 s or only for 10 s (i.e., in the first case, the improvement is minimal, whereas in the second case, it is doubled). A secondary dependent variable was pain perception measured by the differential score between pretest and posttest (differential score = posttest score–pretest score). 

Participants were randomly allocated to the experimental conditions with the sole restriction of maintaining the same proportion of men and women because previous research has shown some gender differences in the performance on the CPT (e.g., [20,21]). Both the participants and the experimenters were blind to the experimental condition to which the participants were allocated. Additionally, we matched the sex of the experimenters and participants because empirical evidence has shown that male participants might show higher tolerance in the presence of a female experimenter than with a male one [22].

### 2.3. Settings and Apparatus

All sessions were conducted individually in an experimental room equipped with a table, two chairs, an armchair, a tablet, headphones, an ice maker machine, and a 30 × 20 × 20 cm glass container. The glass container had two interconnected compartments: one for the ice and the other for the water. In the latter compartment, participants introduced their hands. A digital thermometer was adhered to the container to control the water temperature. Two water pumps (26 L per hour) were also adhered to the glass container to maintain the water circulating. Finally, the ice maker machine was used to keep the temperature of the glass container constant. 

### 2.4. Experimental Task

The CPT was used as the experimental task. The CPT has been broadly used in medical and psychological studies because it produces similar sensations to particular conditions such as chronic pain and persistent psychological distress. Participants were invited to introduce their right hand up to their wrist in a glass container with circulating ice water at 3 to 4 degrees Celsius. They were asked to leave their hands in the water for as long as possible. However, they were also reminded that they were free to remove their hands from the water at any time. Pain tolerance was measured by the total amount of time participants kept their hands in the water. Participants who kept their hands in the water for 300 s at the pretest were excluded from further participation in the study because they reached the maximum admissible pain tolerance according to ethical standards.

### 2.5. Instruments

Generalized Pliance Questionnaire (GPQ; [23]). The GPQ is a measure of generalized pliance consisting of 18 items rated on a 7-point Likert-type scale (7 = always true; 1 = never true). Examples of items are “It is very important for me to be accepted by others,” “My value as a person depends on what other people think and say about me,” or “I care a lot about what my friends think of me.” The GPQ has a one-factor structure and positively correlates with measures of experiential avoidance, cognitive fusion, and emotional symptoms. Conversely, it correlates negatively with valued living and quality-of-life measures. The GPQ obtained an alpha of 0.92 in this study and was used to control the possible influence of social approval on the participants’ performance.

Self-reports of pain during the CPT. After each exposure to the CPT, participants were asked to rate how intense was the induced pain on a 10 cm visual analog scale (VAS). 

### 2.6. Protocols

The protocols were presented in audio through headphones connected to a tablet and lasted for 260 s. They included common components at the beginning and end of the recording. In addition, silences were introduced at the beginning of Protocols B and D in order to control for the duration of the recordings across protocols. In these cases, the audio began with the following instruction: “The audio will start in a few seconds. Please, wait with attention and take this time as a break from the task you have just done.” The metaphor used in this study was based on Sierra et al. [18] and incorporated common physical properties with the discomfort experienced during the CPT and the specification of appetitive augmental functions (i.e., values) to tolerate it. 

Protocol A. This protocol presented the metaphor in the first person and with prompts for relational elaboration (Self and Elaboration). The script of the protocol is presented below:

Introduction: “I am going to make some comments regarding the task you just ended. Your mission during this time is to pay attention to my voice and do the simple imagination exercises that I will suggest to you. First, allow yourself to remember that this experiment aims to analyze what strategies people with chronic or acute pain can use to achieve what is important for them despite experiencing pain. Your participation in this experiment is important because it could contribute to improving the quality of life of these people. We do not expect any result in particular; whatever you do is OK for us. We simply ask you to do the task as naturally as possible and to try to do the following exercise.”

Metaphor: “You have just finished the task with cold water. Please, take a moment to think about the sensation of the cold water on your hand and describe it to yourself (*pause of 15 s*). Imagine you are at the edge of a big swamp. The other side of the swamp is very far away, and it would take you several minutes to get there. On the other side of the swamp, there is the most important thing for you, this thing you dream about, the one that excites you the most and makes you vibrate. *Please, let yourself think for a few seconds about what is on the other side of the swamp and the emotion that drives you to get there (pause of 30 s)*. The swamp’s water is very cold, and when you look at the other side, you realize that the only way to get there is to cross the swamp by swimming. It would take you five minutes to get to the other side. The farther you would swim in the swamp, the colder you would feel, but you would know that you would be much closer to this thing that is so important for you. You would also know that cold is something you would feel momentarily, something uncomfortable that makes sense to feel for a few minutes because on the other side is the most important thing for you. *Please, let yourself imagine the feeling you would have swimming in the swamp while going to the other side and the feeling you would have when seeing the other side closer (pause of 15 s)*. Now, answer yourself the following questions: What would you choose to do? Would you stand at the edge of the swamp watching how the opportunity to move closer to what is most important for you fades away on the other side, or would you jump into the water and swim despite the discomfort of the cold? *(pause of 10 s*).”

Closing the protocol: “Now, you are going to do the task again. We suggest you try to put into practice what the story told you and see if it could help you bear the task’s discomfort better. Remember that anything you do is OK for us and that we are not expecting anything special in any direction. We have finished. Please call the experimenter.”

Protocol B. This protocol was the same as Protocol A, but in point 2, there were no pauses and cues to prompt relational elaboration (italicized parts; Self and No Elaboration). 

Protocol C. This protocol was the same as Protocol A but substituting in point 2 the word “you” (words highlighted) for “Paolo” or “Paola” depending on the participant’s sex (Other and Elaboration). 

Protocol D. This protocol was the same as Protocol A, but in point 2, the word “you” was substituted by “Paolo” or “Paola,” and there were no pauses and cues to prompt for relational elaboration (Other and No Elaboration).

### 2.7. Procedure

This study was approved by The Bioethics Committee of Fundación Universitaria Konrad Lorenz (2016-021B). All participants in this study provided written informed consents. The experimental sessions were conducted individually and lasted for approximately 30 min, distributed in four phases (see Figure 1).

*Phase 1.* Pre-experimental measures. Participants provided written informed consent for their participation in the study. Participants were asked to report in the informed consent if they were suffering from some medical history incompatible with the CPT (see exclusion criteria in the participants’ section). In order to make the experimental task valuable, participants were told that the study aimed to analyze what kind of coping strategies might be helpful to people suffering from constant pain or who have to deal with situations that are accompanied by high discomfort. Then, participants responded to the GPQ.

*Phase 2.* Pretest of the CPT. Participants were first exposed to the CPT in the presence of the experimenter. Participants received the following instruction: “Insert your right hand up to the wrist and keep it in as long as possible. Remember that you can take it out at any time.” The experimenter measured with a chronometer the time since the participant inserted the hand until they removed it. Participants were asked to respond to the VAS of the perceived pain after removing the hand.

*Phase 3.* Protocols. Participants were then asked to sit on the armchair, put on the headphones connected to the tablet, and push the play button of the audio player app. The audio player was programmed to randomly reproduce one of the four protocols. This way, the participants were randomly assigned to one of the four experimental conditions. To avoid the potential influence of expectations, the experimenter abandoned the experimental room during the reproduction of the audio file. Thus, the experimenter did not know which experimental protocol was listened to by the participant (i.e., the experimenter was blinded). 

*Phase 4.* Posttest of the CPT. After listening to the randomly assigned protocol, participants informed the experimenter that the audio file had finished. Then, the experimenter invited participants to perform the CPT again. Afterward, they were completely debriefed about the aims of the experiment.

### 2.8. Data Analysis

Bayesian informative analyses of variance (BAIN ANOVAs) were computed on the free software JASP 0.14.1.0. This software provides a graphical interface to several R packages, including BAIN [24,25,26,27]. BAIN ANOVA permits evaluating informative hypotheses using the Bayes factor. The Bayes factor (BF) is a ratio that quantifies how much the observed data support two competing hypotheses. When BF_12_ = 1, there is no preference for either hypothesis 1 (H_1_) or hypothesis 2 (H_2_). However, BF_12_ > 1 indicates that H_1_ should be preferred; conversely, BF_12_ < 1 indicates that H_2_ should be preferred. Bayes factors are usually interpreted according to Jeffreys [28] and Wagenmakers, Wetzels, Borsboom, and van der Maas [29]: 1 = no evidence; 1–3 = anecdotal evidence for H_1_; 3–10 = substantial evidence for H_1_; 10–30 = strong evidence for H_1_; 30–100 = very strong evidence for H_1_; and >100 = extreme evidence for H_1_ (note that BF_12_ < 1 are interpreted in the same way, but favoring H_2_).

The BAIN framework has significant advantages over the typical frequentist ANOVA based on null hypothesis significance testing (NHST) [30]. Contrary to the NHST, the BAIN framework takes into account the evidence favoring each of the hypotheses under consideration, including the null hypothesis. Thus, the BAIN framework overcomes the limitation of making dichotomous reject/do-not-reject decisions typical of the NHST [30]. Another important advantage is that the BAIN framework allows the researcher to test informative hypotheses [27], such as mA > mB = mC = mD or mA = mC > mB = mD, where mA, mB, mC, and mD denote the means in four experimental conditions (i.e., Conditions A to D). Finally, the BAIN framework allows evaluating multiple alternative hypotheses with the same standing and without having to account for multiple testing [27]. According to these advantages of the BAIN framework, we adopted it to test the support obtained by the different theoretically coherent hypotheses (see below). 

As traditional ANOVA, the BAIN ANOVA is also sensitive to outliers and the violation of model assumptions [31]. In this study, however, the assumption of homoscedasticity was not considered because it seems irrelevant when the groups have equal size, and BAIN ANOVA appears to be robust to its violations [27,32,33]. The performance on the CPT usually presents outliers (e.g., [18]). Thus, the presence of outliers in the percentage of time tolerating the pain at the posttest in relation to the pretest was analyzed graphically for each experimental condition. A total of 1, 4, 0, and 3 outliers were found for Condition A to D, respectively. These outliers were replaced with the next higher value following the Winsor method [34]. 

Firstly, we explored the equivalence of generalized pliance, pretest tolerance, and pretest pain intensity across the experimental conditions by computing BAIN ANOVAs. In so doing, we compared two hypotheses: a hypothesis in which all means are the same (H_1_: mA = mB = mC = mD) and a hypothesis in which the means are unrestricted (H_u_: mA, mB, mC, mD). This second hypothesis is equivalent to the alternative hypothesis in the frequentist ANOVA. Secondly, the differential scores on pain intensity across conditions were also analyzed with these two hypotheses because previous research has shown that this type of protocol does not alter pain intensity differentially (e.g., [18,19,35]). 

Thirdly, we analyzed the data on the increase of pain tolerance by defining six alternative hypotheses. The first hypothesis reflects our expectations that participants in Condition 1 would show a higher increase in pain tolerance than the remaining conditions, which would not show differences among them (H_1_: mA > mB = mC = mD). In other words, this hypothesis indicates that the components Self and Elaboration do not have an effect separated, but they show a positive interaction in increasing pain tolerance. The second hypothesis indicates that both components (i.e., Self and Elaboration) have a positive effect on pain tolerance and that they also interact positively (H_2_: mA > mB = mC > mD). The third hypothesis suggests that both Self and Elaboration positively affect pain tolerance, but they do not interact (H_3_: mA = mB = mC > mD). The fourth hypothesis implies that only the component Self positively affects pain tolerance (H_4_: mA = mB > mC = mD). The fifth hypothesis implies that only the component Elaboration positively affects pain tolerance (H_5_: mA = mC > mB = mD). Lastly, the sixth hypothesis is equivalent to the null hypothesis that Self and Elaboration do not affect pain tolerance (H_6_: mA = mB = mC = mD). In all BAIN ANOVAs, we used the default prior distribution suggested in BAIN, which gives all hypotheses the same likelihood and uses a fraction of the information in the data to specify the variance of the prior distribution. Lastly, between-condition effect sizes in Phase 4 were calculated with Cohen’s *d* [36], which can be interpreted as small (*d* = 0.20 to 0.49), medium (*d* = 0.50 to 0.79), and large (above *d* = 0.80).

## 3. Results

### 3.1. Initial Equivalence between Groups 

Four participants kept their hands in the water for 300 s at the pretest. Accordingly, they were excluded from further participation in the study because they reached the maximum admissible pain tolerance according to ethical standards. Table 1 shows that the one-way BAIN ANOVAs revealed that the hypothesis of no differences across conditions (i.e., H_1_) was supported over the hypothesis in which the means are unrestricted (H_u_) for generalized pliance (BF_1u_ = 70.426), pretest tolerance (BF_1u_ = 9.870), and pain intensity (BF_1u_ = 18.608). Accordingly, the experimental conditions seem to be equivalent at pretest on these variables.

### 3.2. Effect of the Experimental Protocols

Participants’ performance on pain tolerance for each experimental condition can be observed in Figure 2. A total of 10 out of 20 participants in Condition A showed improvements in pain tolerance greater than 250%, 6 in Condition B, 5 in Condition C, and 4 in Condition D.

Figure 3 and Table 2 show the descriptive data in the increase of pain tolerance from pretest to posttest for each experimental condition. Again, participants in Condition A showed the highest score, whereas Conditions B, C, and D showed similar results. 

Table 3 shows the results of the one-way BAIN ANOVA analyzing the pain tolerance increase from pretest to posttest. All hypotheses were given the same a priori model probability (i.e., *MP* = 0.167). After observing the data, the hypothesis with the highest posterior model probability was H_1_ (*PMP* = 0.545), which implies that Self and Elaboration showed a positive interaction on pain tolerance but did not have an effect separately. The second hypothesis with the highest PMP was H_2_ (*PMP* = 0.223), which suggests that both Self and Elaboration positively affected pain tolerance and that they interacted. The remaining hypotheses showed a decrease from *MP* to *PMP*. Taking together H_1_, H_2_, and H_3_, the data of this study indicate that both Self and Elaboration had some type of positive effect on pain tolerance with a *PMP* near 0.80 (*PMP* = 0.796). Finally, it is worth noting that the null hypothesis indicating that Self and Elaboration did not have any positive effect on pain tolerance (i.e., H_6_) showed a very low posterior probability (*PMP* = 0.025). 

Table 3 also presents the BF matrix comparing all pairs of hypotheses. The observed data provided strong evidence supporting H_1_ over H_3_ (BF_13_ = 19.773), H_4_ (BF_14_ = 23.103), and H_6_ (BF_16_ = 21.520). In addition, the data provided substantial evidence supporting H_1_ over H_5_ (BF_15_ = 3.510), but only anecdotal evidence supporting H_1_ over H_2_ (BF_12_ = 2.439). The effect sizes between Condition A and the remaining conditions were in the range between medium to large (Condition A vs Condition B: *d* = 0.73; Condition A vs. Condition C: *d* = 0.47; Condition A vs. Condition D: *d* = 0.76). 

Table 2 also shows the descriptive data concerning differential pain intensity for each condition. The one-way BAIN ANOVA indicates that, after observing the data, the hypothesis with the highest *PMP* was H_1_ (*PMP* = 0.991), which implies no differences in changes in pain intensity across conditions. The hypothesis in which the means are unrestricted obtained a minimal posterior probability (*PMP* = 0.009). The BF showed extreme support for the H_1_ over the H_u_ (BF1u = 110.111). 

## 4. Discussion

The experiential delivery of metaphors has been encouraged in different psychotherapies. However, little research has been conducted regarding the variables that enhance the effect of metaphors. One advantage of ACT is its closed relation with the functional-contextual approach of language and cognition called RFT. Based on RFT analyses [17], previous experimental analogs found that including common physical properties and explicit appetitive augmental functions enhanced the efficacy of metaphors [18,19]. This study represents a continuation of this research line by analyzing two additional components that seemed promising according to the RFT analyses of perspective-taking and the transformation of functions through the rules that can be derived with the introduction of metaphors. The first component was presenting the metaphor by asking the individual to imagine herself as the story’s protagonist versus using a fictitious character (Self vs. Other). The second component was including versus not including relational cues prompting the relational elaboration of the rules that might be derived from the metaphor (Elaboration vs. No Elaboration). The current study aimed to analyze the effect of these two components involved in the experiential delivery of metaphors in a double-blind, randomized, 2 × 2 factorial experiment. 

The results from the one-way BAIN ANOVAs supported the hypotheses of no differences across conditions in generalized pliance, pretest pain tolerance, and pretest pain intensity. Therefore, the experimental conditions were equivalent in these variables before the introduction of the intervention protocols. Regarding the pre–post change in pain tolerance, we compared six alternative hypotheses: H_1_ = Self and Elaboration have no effect separated, but they show a positive interaction; H_2_ = Both Self and Elaboration have a positive effect, and they interact; H_3_ = Both Self and Elaboration have a positive effect, but they do not interact; H_4_ = Self has a positive effect, but Elaboration not; H_5_ = Elaboration has a positive effect, but Self not; and H_6_ = Self and Elaboration do not have a positive effect of any kind. The one-way BAIN ANOVA indicated that the first hypothesis was the most supported by the observed data with a *PMP* = 0.545, followed by H_2_ with a *PMP* = 0.223. The remaining hypotheses obtained *PMP*s lower than the prior model probability. In addition, the BFs indicated that the H_1_ obtained from substantial to strong evidence against H_3_ to H_6_, but only anecdotal evidence against H_2_. 

Our initial hypothesis of Self and Elaboration only showing a positive interaction effect on pain tolerance (i.e., H_1_) received the greatest support by the observed data. However, the hypothesis of Self and Elaboration having a positive effect separately and an interaction effect (i.e., H_2_) should not be discarded yet according to the associated PMP and the BF_12_ value. Further research might replicate the current findings and provide additional information regarding the relative fit of H_1_ versus H_2_, and potential moderator variables affecting both the effect of Self and Elaboration alone and in interaction. 

As previous similar studies did not find differential pre–post changes in pain intensity across experimental conditions (e.g., [18,19,35]), we did not expect a differential change in this study either. The observed data strongly supported this hypothesis compared to the hypothesis in which the means were unrestricted. This result suggests that the process of change of the metaphor’s components analyzed in this study was not the decrease of pain intensity but the reduction of the discriminative functions for avoiding pain and the contact with powerful symbolic reinforcing consequences for approaching it. In other words, the inclusion of Self and Elaboration seemed to positively affect pain tolerance by promoting an experiential contact with the reinforcing consequences established by the rules derived from the metaphor content. 

Some limitations of the study are worth mentioning. Firstly, the effect of the protocols was tested only in one experimental task. Further research might be conducted with alternative experimental tasks to allow a better generalization of the results. This is particularly important to draw conclusions for applications outside laboratory experiments (e.g., clinical metaphors in psychotherapy). In this regard, the results of this study can be more easily extrapolated to clients suffering from chronic or acute pain. Secondly, only undergraduate students participated in this study, which reduces the generalizability of the results. Thirdly, the effect of the independent variables was analyzed within the context of a metaphor including two known active components: common physical properties and augmental functions [18,19]. Thus, it is unclear whether, in the context of a metaphor without these active components, being the protagonist of the metaphor and providing cues prompting relational elaboration would have the same effect. For instance, both variables might not have a significant effect if the metaphor lacks a valued context. Further research might investigate whether certain conditions are required for these two components to have a significant effect. 

Fourthly, some features of this experimental analog might not correspond with usual clinical interactions. For instance, the therapists usually introduce the metaphors instead of playing an audio file. Conversely, this experimental analog would be more similar to the introduction of metaphors in digital interventions. Lastly, the study did not explore whether the skills in analogical reasoning and perspective-taking moderated the effect of the protocols. For instance, it could be that participants in Condition D with high analogical and perspective-taking skills would show a pain tolerance similar to participants in Condition A. Future studies might include measures of these skills to conduct moderation analyses. 

In summary, this study suggests that introducing metaphors with the client as the protagonist of the story (i.e., Self) and providing prompts for the relational elaboration of the rules derived from the metaphor content (i.e., Relational Elaboration) might increase the effect of the metaphor. These components seem to increase the metaphor effect mainly when introduced together (i.e., they have a positive interaction effect), which is consistent with our initial hypothesis. Specifically, prompting relational elaboration makes sense mainly in the context of being the protagonist because this allows experientially contacting the symbolic and emotional consequences derived from the metaphor content. In other words, relational elaboration, in this case, would permit deriving and following track and augmental rules related to own values. Conversely, when the metaphor is introduced with a fictitious protagonist, contacting personally relevant symbolic consequences would require a perspective-taking step. Thus, the probability of relational elaboration increasing the metaphor effect would depend on the client’s perspective-taking skills. 

## 5. Conclusions

In conclusion, this study continued a research line that aims to develop and test RFT conceptualizations of clinical processes. If further research confirms and extends this study’s results, therapists might be trained to introduce metaphors with the client as the protagonist and prompt relational elaboration of the rules derived from the metaphor content. These suggestions are in line with theoretical RFT texts that provide guidelines on the use of experiential metaphors [3,4,6,11]. 

## Figures and Tables

**Figure 1 ijerph-18-10630-f001:**
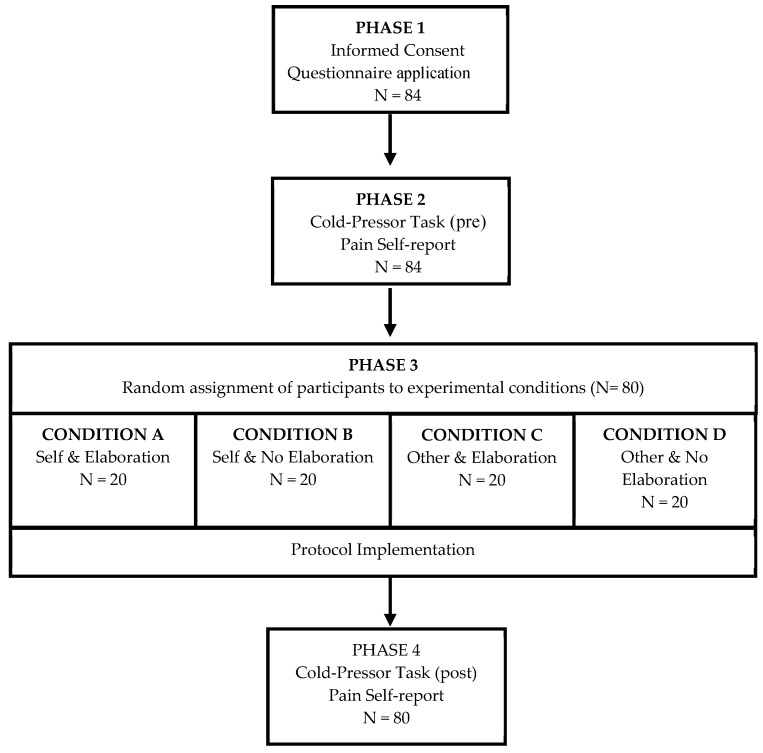
Overview of the experimental procedure.

**Figure 2 ijerph-18-10630-f002:**
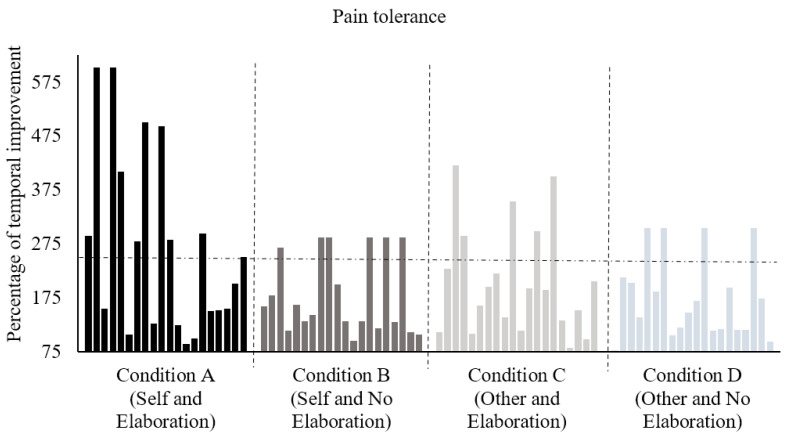
Percentage of temporal improvement in pain tolerance in posttest in relation to pretest for participants of each experimental condition. The horizontal, dashed line highlights improvements above 250% of temporal improvement.

**Figure 3 ijerph-18-10630-f003:**
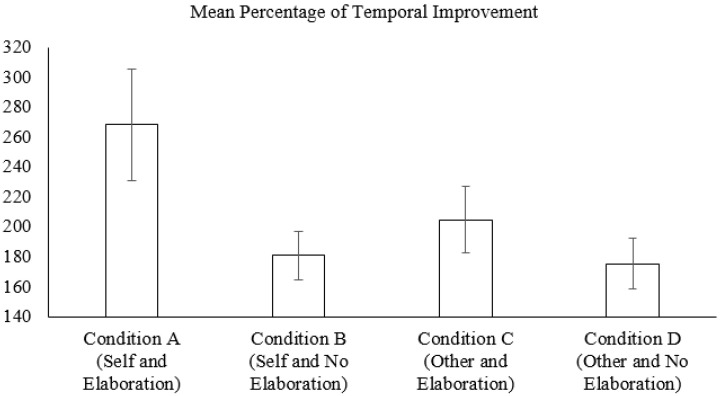
Mean percentage of temporal improvement in posttest in relation to pretest for each experimental condition and error bars.

**Table 1 ijerph-18-10630-t001:** Descriptive data for each condition at pretest and BAIN one-way ANOVA results.

	Condition A M (SD)	Condition B M (SD)	Condition C M (SD)	Condition D M (SD)	PMP H_1_	PMP H_u_	BF_1u_
Generalized Pliance	56.90 (21.37)	52.15 (15.52)	50.35 (19.67)	52.95 (19.05)	0.986	0.014	70.426
Pretest Pain Tolerance	34.95 (30.71)	51.20 (71.45)	66.00 (67.05)	32.80 (20.27)	0.908	0.092	9.870
Pretest Pain Intensity	6.60 (1.66)	6.98 (1.77)	6.57 (1.44)	7.46 (1.46)	0.949	0.051	18.608

Note. BF_1u_ = Bayes factor of the H_1_ versus the H_u_; H_1_: mA = mB = mC = mD; H_u_: mA, mB, mC, mD; PMP = posterior model probabilities.

**Table 2 ijerph-18-10630-t002:** Descriptive data for each condition regarding the percentage time tolerating pain at posttest in relation to pretest and pre–post change in pain intensity.

	Pain Tolerance				Pain Intensity			
			95% CI				95% CI	
	M	SD	Lower	Upper	M	SD	Lower	Upper
Condition A (Self & Elaboration)	268.211	167.467	219.610	316.812	−0.920	2.358	−1.955	0.115
Condition B (Self & No Elaboration)	180.857	73.013	132.256	229.458	−0.763	2.822	−1.825	0.299
Condition C (Other & Elaboration)	204.809	100.193	156.209	253.410	−0.645	2.431	−1.680	0.390
Condition D (Other & No Elaboration)	175.407	75.998	126.806	224.008	−0.511	1.689	−1.572	0.551

Note. CI = credible interval.

**Table 3 ijerph-18-10630-t003:** Posterior model probabilities and Bayes factors matrix of the alternative hypotheses evaluated regarding the increase in pain tolerance. The Bayes factor matrix compares the hypothesis on the row against the hypothesis on the file.

	Bayes Factors						
Hypothesis	MP	PMP	H_2_	H_3_	H_4_	H_5_	H_6_
H_1_: mA > mB = mC = mD. Self & Elaboration have no effect separated, but they interact.	0.167	0.545	2.439	19.773	23.103	3.510	21.520
H_2_: mA > mB = mC > mD. Both Self & Elaboration have an effect, and they interact.	0.167	0.223	1.000	8.108	9.474	1.439	8.824
H_3_: mA = mB = mC > mD. Both Self & Elaboration have an effect, but they do not interact.	0.167	0.028	0.123	1.000	1.168	0.178	1.088
H_4_: mA = mB > mC = mD. Self has an effect, but Elaboration not.	0.167	0.024	0.106	0.856	1.000	0.152	0.931
H_5_: mA = mC > mB = mD. Elaboration has an effect, but Self not.	0.167	0.155	0.695	5.634	6.582	1.000	6.131
H_6_: mA = mB = mC = mD. Self & Elaboration have no effect separated, nor they interact.	0.167	0.025	0.113	0.919	1.074	0.163	1.000

Note. *MP* = prior model probabilities; *PMP* = posterior model probabilities.

## Data Availability

The data presented in this study are available on request from the corresponding author. The data are not publicly available for privacy reasons.

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
