# Peer review of "Empirical Investigation of the Verbal Cues Involved in Delivering Experiential Metaphors"

_ijerph, 2021, doi:10.3390/ijerph182010630_

Round 1

Reviewer 1 Report

Abstract
The abstract is not comprehensible. The background, the aim of the study, and the results are not reported. The abstract should be offered an overview intelligible and helpfulness to understanding the study. I suggest to the authors organize the abstract, including in background, aim, methods, results, and conclusions. In particular, I recommend reporting the main results of the analysis in the abstract and to review by attention the methods (the conditions C and D are reported as equivalent).

Introduction
The introduction is too long. Although, it‘s appreciable by authors the the will to provide a deeply overview of the topic, I suggest synthesizing the introduction, highlight more possible the aim of the study and the necessity of this research. I suggest reporting the design of the study in the methods. 

Methods
Since, the presented work, is a randomized study, the study should be conducted and reported following the CONSORT checklist. 
The description of the method is uncompleted. The description of the sample is partial; only female subjects are reported. I suggest reporting the descriptive all characteristics (e.g. gender, age, educational) of the sample in the detailed table. 
Some information about the recruitment phase is needed: how the sample has been recruited (they were college’s students?); which were the inclusion criteria? Since, the outcome was the measurement of pain, the presence of psychiatric or neurological conditions were evaluated in the recruited sample?
The study was registered in the Clinical trials.gov? Since the study is a randomized study, the flow diagram might be reported.
The measurement of pain was the principal outcome? The description of randomized procedures should be reported. The experimenters that conducted the experiment conducted also the assessment?
The description of the treatment protocol is uncompleted. The treatment was administrated during an ACT therapy or some psychotherapy intervention? The authors could provide a more detailed description of the setting of the protocol.
Results
The subjects showed differences at the baseline for the principal outcome?
The authors could explain why the Bayesian informative analyses of variance were used instead linear model?
Discussion
The authors should provide more references to other studies to support the results. I appreciated that limits are described by authors, but the novelty of results are not highlighted in the discussion and is not clear to me which improvement to the topic the present study provieded.

Author Response

We would like to thank the comments and suggestions made by all reviewers. Please, see below our response to the suggestions and a detailed inform of the changes introduced in the manuscript.

Reviewer #1

Comment 1: The abstract is not comprehensible. The background, the aim of the study, and the results are not reported. The abstract should be offered an overview intelligible and helpfulness to understanding the study. I suggest to the authors organize the abstract, including in background, aim, methods, results, and conclusions. In particular, I recommend reporting the main results of the analysis in the abstract and to review by attention the methods (the conditions C and D are reported as equivalent).

Response to Comment 1: Thanks for identifying that the description of Conditions C and D were the same. We have amended that. We have also incorporated the information required in the abstract.

Comment 2: The introduction is too long. Although, it‘s appreciable by authors the will to provide a deeply overview of the topic, I suggest synthesizing the introduction, highlight more possible the aim of the study and the necessity of this research. I suggest reporting the design of the study in the methods.

Response to Comment 2: Thanks for the comment. We acknowledge that the introduction might be unusually long, but this extension allowed us to conceptualize the experiment in basic terms. Accordingly, we chose not synthesizing the introduction because that would lead to losing contact with important parts of theoretical background and conceptualization of this study. Conversely, we also acknowledge that the last paragraph of the introduction is also presented when explaining the design of the experiment. Therefore, we have followed the suggestion of eliminating it.

Comment 3: Since, the presented work, is a randomized study, the study should be conducted and reported following the CONSORT checklist.

Response to Comment 3: We have the impression that the reviewer thinks that we reported a randomized clinical trial (see also Comment 6). However, the manuscript reports an experimental analog.

Comment 4: The description of the method is uncompleted. The description of the sample is partial; only female subjects are reported. I suggest reporting the descriptive all characteristics (e.g. gender, age, educational) of the sample in the detailed table.

Response to Comment 4: We have included that there were 42 men (“Eighty-four undergraduates (42 women and 42 men; age range = 18 to 40; M = 19.64, SD = 2.75), attending different courses in a university in Bogotá (Colombia), participated in the experiment”). As can be seen, we also provided the age range, and its descriptive data (mean and standard deviation). All participants were undergraduates.

Comment 5: Some information about the recruitment phase is needed: how the sample has been recruited (they were college’s students?); which were the inclusion criteria? Since, the outcome was the measurement of pain, the presence of psychiatric or neurological conditions were evaluated in the recruited sample?

Response to Comment 5: The participants section states that all participants were undergraduates, and the exclusion criteria were (a) having previous experience with the procedures (i.e., the CPT) or the theory involved in this study (i.e., RFT and ACT), and (b) suffering from cardiac and circulatory affections, hypertension, diabetes, epilepsy, chronic pain conditions, or recent wounds. These exclusion criteria were established the experimental task might adversely affect participants suffering from these conditions. We have added information about the inclusion criteria: “The inclusion criteria were (a) being an undergraduate in the university in which the study was conducted (the Institutional Bioethical Committee asked this to ensure that participants were protected by the institutional insurance in case of accident or an unlikely adverse reaction to the experimental procedures), and (b) being equal or older than 18 years.”

Comment 6: The study was registered in the Clinical trials.gov? Since the study is a randomized study, the flow diagram might be reported.

Response to Comment 6: It seems the reader understood that the manuscript reports a randomized clinical trial. However, we report an experimental analog with nonclinical participants. The intervention protocols did not aim to improve clinical characteristics of the sample.

Comment 7: The measurement of pain was the principal outcome?

Response to Comment 7: The Design and Variables section includes statements about the primary dependent variable (“The primary dependent variable was pain tolerance…”) and the secondary dependent variable (“A secondary dependent variable was pain perception…).

Comment 8: The description of randomized procedures should be reported.

Response to Comment 8: Thanks for noting that we did not specify how the randomization was conducted. We have tried to clarify this and other relevant aspects of the procedures in Phases 3 and 4. The text now says:

Phase 3. Protocols. Participants were then asked to sit on the armchair, put on the headphones connected to the tablet, and push the play button of the audio player app. The audio player was programmed to randomly reproduce one of the four protocols. This way, the participants were randomly assigned to one of the four experimental conditions. To avoid the potential influence of expectations, the experimenter abandoned the experimental room during the reproduction of the audio file. Thus, the experimenter did not know which experimental protocol was listened by the participant (i.e., the experimenter was blinded).

Phase 4. Posttest of the CPT. After listening to the randomly assigned protocol, participants informed the experimenter that they audio file had finished. Then, the experimenter invited participants to perform the CPT again. Afterward, they were completely debriefed about the aims of the experiment.

Comment 9: The experimenters that conducted the experiment conducted also the assessment?

Response to Comment 9: Yes, as can be seen in the procedure section, only one experimenter administered the experimental procedures for each participant. However, the experimenter was blind to the intervention protocol randomly assigned to the participant.

Comment 10: The description of the treatment protocol is uncompleted. The treatment was administrated during an ACT therapy or some psychotherapy intervention? The authors could provide a more detailed description of the setting of the protocol.

Response to Comment 10: Please, see response to Comment 6. We did not conduct a randomized clinical trial. The protocol in Condition A is completely scripted. We do not present the text of the remaining protocols because they were almost the same with the exceptions mentioned in the text.

Comment 11: The subjects showed differences at the baseline for the principal outcome?

Response to Comment 11: The first part of the results section mention that all experimental conditions were equivalent in all variables measured in the pre-test: “Table 1 shows that the one-way BAIN ANOVAs revealed that the hypothesis of no differences across conditions (i.e., H1) was supported over the hypothesis in which the means are unrestricted (Hu) for generalized pliance (BF1u = 70.426), pretest tolerance (BF1u = 9.870), and pain intensity (BF1u = 18.608). Accordingly, the experimental conditions seem to be equivalent at pretest on these variables.”

Comment 12: The authors could explain why the Bayesian informative analyses of variance were used instead linear model?

Response to Comment 12: The advantages of the BAIN framework compared to the typical frequentist ANOVA framework were explained in the text in the following paragraph: “The BAIN framework has significant advantages over the typical frequentist ANOVA based on null hypothesis significance testing (NHST) [31]. Contrary to the NHST, the BAIN framework takes into account the evidence favoring each of the hypotheses under consideration, including the null hypothesis. Thus, the BAIN frame-work overcomes the limitation of making dichotomous reject/do-not-reject decisions typical of the NHST [31]. Another important advantage is that the BAIN framework allows the researcher to test informative hypotheses [28], such as mA > mB = mC = mD or mA = mC > mB = mD, where mA, mB, mC, and mD denote the means in four experimental conditions (i.e., Conditions A to D). Finally, the BAIN framework allows evaluating multiple alternative hypotheses with the same standing and without having to account for multiple testing.”

According to the reviewer’s suggestion, we have added the following sentence: “According to these advantages of the BAIN framework, we adopted it to test the sup-port obtained by the different theoretically coherent hypotheses (see below).”

Comment 13: The authors should provide more references to other studies to support the results. I appreciated that limits are described by authors, but the novelty of results are not highlighted in the discussion and is not clear to me which improvement to the topic the present study provided.

Response to Comment 13: Thanks for the comment. We have extended the discussion section to include more explicit references to previous studies.

Reviewer 2 Report

Good research design and excellent data analysis. 
I suggest checking something:

1) Figure 2 seems to be a bit wrong compared to text at lines 415-416. The horizontal dotted line indicating 250% slightly below when compared with the text indicating 10, 6, 4, and 4, while if we consider the bars that exceed the line, we would have 9, 6, 5, 4. Better to check: it could be a typographical problem or an error in the graph.

2) The Bayesian approach, although not recent, is little known and has only been spreading in recent years. The description on lines 352-362 is good and acceptable. 
However, it is not clear whether the Bayesian approach allows the model's assumptions to be ignored. On lines 363-365, the authors cite Field et al. and Zimmerman [references 33,34] to justify the irrelevance of the problem. However, both references have to do with the NHST approach and not with the Bayesian approach. In particular, [34] concerns only the t-test and not the Anova. 

Author Response

Reviewer #2

Comment 1: Good research design and excellent data analysis.

Response to Comment 1: Thanks for the nice comments.

Comment 2: Figure 2 seems to be a bit wrong compared to text at lines 415-416. The horizontal dotted line indicating 250% slightly below when compared with the text indicating 10, 6, 4, and 4, while if we consider the bars that exceed the line, we would have 9, 6, 5, 4. Better to check: it could be a typographical problem or an error in the graph.

Response to Comment 2: Thanks for noting the mistakes. We have amended the figure so that the horizontal dashed line coincides with the value of 250. We have also corrected the mistake in the number of participants in condition C that exceeded the line.

Comment 3: The Bayesian approach, although not recent, is little known and has only been spreading in recent years. The description on lines 352-362 is good and acceptable. However, it is not clear whether the Bayesian approach allows the model's assumptions to be ignored. On lines 363-365, the authors cite Field et al. and Zimmerman [references 33,34] to justify the irrelevance of the problem. However, both references have to do with the NHST approach and not with the Bayesian approach. In particular, [34] concerns only the t-test and not the Anova.

Response to Comment 3: Thanks for the comment and concern. Frequentist Student’s t and ANOVA are considered generally robust to violations of the homoscedasticity assumption when group sizes are equal or approximately equal. We included the reference to Zimerman (2004) because the author explicitly recommended not testing for the homocedasticity assumption. His study focused on the t-test, but these two tests are intrinsically related because ANOVA expands the basic concepts developed in the t-test. Although there is less literature concerning assumption violations in BAIN ANOVA, it is considered that the concerns regarding them are similar to frequentist ANOVA (Hoijtink, Gu, & Mulder, 2019; Hoijtink, Mulder, van Lissa, & Gu, 2019). For instance, Hoijtink, Mulder, et al. (2019) commented that: “There is one study in the context of ANOVA into the effect of violation of the assumption of homogeneous variances on hypotheses evaluation by means of the Bayes factor (Van Rossum, van de Schoot, & Hoijtink, 2013). Although further study is definitely needed, it appears that the Bayes factor, like NHST, is robust if the violations.” (p. 11). We have included this reference in the sentence and included the following: “In this study, however, the assumption of homoscedasticity was not considered because it seems irrelevant when the groups have equal size and BAIN ANOVA appears to be robust to its violations [28, 33, 34].”

Reviewer 3 Report

Thank you for this interesting experiment which you explain clearly and present well in this paper.  I would have appreciated a little more location of yourselves as researchers and the context of the sample that you used including geography and any relevant demographic issues which may be helpful in interpreting your approach and findings.  You chose a helpful scenario which could be adaptable and transferable by specialist readers into their context.  

Author Response

Comment 1: Thank you for this interesting experiment which you explain clearly and present well in this paper. I would have appreciated a little more location of yourselves as researchers and the context of the sample that you used including geography and any relevant demographic issues which may be helpful in interpreting your approach and findings. You chose a helpful scenario which could be adaptable and transferable by specialist readers into their context. 

Response to Comment 1: Thanks for the nice comments and suggestions. We have added in the participants section that the participants were attending a university in Bogotá (Colombia). We have not specified data regarding race/ethnicity because the population in Bogotá, and Colombia in general, is predominantly composed by mestizos and white people, without establishing clear differences among them.

Round 2

Reviewer 1 Report

The manuscript has been improved by the authors